# A Zero-Positive Learning Approach for Diagnosing Software Performance Regressions

**Mejbah Alam**
Intel Labs
mejbah.alam@intel.com

**Justin Gottschlich**
Intel Labs
justin.gottschlich@intel.com

**Nesime Tatbul**
Intel Labs and MIT
tatbul@csail.mit.edu

**Javier Turek**
Intel Labs
javier.turek@intel.com

**Timothy Mattson**
Intel Labs
timothy.g.mattson@intel.com

**Abdullah Muzahid**
Texas A&M University
abdullah.muzahid@tamu.edu

## Abstract

The field of *machine programming* (MP), the automation of the development of software, is making notable research advances. This is, in part, due to the emergence of a wide range of novel techniques in machine learning. In this paper, we apply MP to the automation of software performance regression testing. A *performance regression* is a software performance degradation caused by a code change. We present *AutoPerf* – a novel approach to automate regression testing that utilizes three core techniques: *(i)* zero-positive learning, *(ii)* autoencoders, and *(iii)* hardware telemetry. We demonstrate AutoPerf's generality and efficacy against 3 types of performance regressions across 10 real performance bugs in 7 benchmark and open-source programs. On average, AutoPerf exhibits 4% profiling overhead and accurately diagnoses more performance bugs than prior state-of-the-art approaches. Thus far, AutoPerf has produced no false negatives.

## 1 Introduction

*Machine programming* (MP) is the automation of the development and maintenance of software. Research in MP is making considerable advances, in part, due to the emergence of a wide range of novel techniques in machine learning and formal program synthesis [11, 12, 16, 37, 43, 44, 46, 47, 52, 55, 62]. A recent review paper proposed *Three Pillars of Machine Programming* as a framework for organizing research on MP [25]. These pillars are *intention*, *invention*, and *adaptation*.

*Intention* is concerned with simplifying and broadening the way a user's ideas are expressed to machines. *Invention* is the exploration of ways to automatically discover the right algorithms to fulfill those ideas. *Adaptation* is the refinement of those algorithms to function correctly, efficiently, and securely for a specific software and hardware ecosystem. In this paper, we apply MP to the automation of software testing, with a specific emphasis on parallel program performance regressions. Using the three pillars nomenclature, this work falls principally in the adaptation pillar.

*Software performance regressions* are defects that are erroneously introduced into software as it evolves from one version to the next. While they do not impact the functional correctness of the software, they can cause significant degradation in execution speed and resource efficiency (e.g., cache contention). From database systems to search engines to compilers, performance regressions are

commonly experienced by almost all large-scale software systems during their continuous evolution and deployment life cycle [7, 24, 30, 32, 34]. It may be impossible to entirely avoid performance regressions during software development, but with proper testing and diagnostic tools, the likelihood for such defects to silently leak into production code might be minimized.

Today, many benchmarks and testing tools are available to detect the presence of performance regressions [1, 6, 8, 17, 42, 57], but diagnosing their root causes still remains a challenge. Existing solutions either focus on whole program analysis rather than code changes [15], or depend on previously seen instances of performance regressions (i.e., rule-based or supervised learning approaches [20, 29, 33, 59]). Furthermore, analyzing multi-threaded programs running over highly parallel hardware is much harder due to the *probe effect* often incurred by traditional software profilers and debuggers [23, 26, 27]. Therefore, a more general, lightweight, and reliable approach is needed.

In this work, we propose AutoPerf, a new framework for software performance regression diagnostics, which fuses multiple state-of-the-art techniques from hardware telemetry and machine learning to create a unique solution to the problem. First, we leverage *hardware performance counters (HWPCs)* to collect fine-grained information about run-time executions of parallel programs in a lightweight manner [10]. We then utilize *zero-positive learning (ZPL)* [36], *autoencoder neural networks* [60], and *k-means clustering* [35] to build a general and practical tool based on this data. Our tool, AutoPerf, can learn to diagnose potentially any type of regression that can be captured by HWPCs, with minimal supervision.

We treat performance defects as anomalies that represent deviations from the normal behavior of a software program. Given two consecutive versions of a program $P$, $P_i$ and $P_{i+1}$, the main task is to identify anomalies in $P_{i+1}$'s behavior with respect to the normal behavior represented by that of $P_i$. To achieve this, first we collect HWPC profiles for functions that differ in $P_i$ and $P_{i+1}$, by running each program with a set of test inputs. We then train autoencoder models using the profiles collected for $P_i$, which we test against the HWPC profiles collected for $P_{i+1}$. Run instances where the autoencoder reconstruction error (RE) is above a certain threshold are classified as regressions. Finally, these regressions are analyzed to determine their types, causes, and locations in $P_{i+1}$.

Our framework enhances the state of the art along three dimensions:

- *Generality:* ZPL and autoencoders eliminate the need for labeled training data, while HWPCs provide data on any detectable event. This enables our solution to generalize to any regression pattern.
- *Scalability:* Low-overhead HWPCs are collected only for changed code, while training granularity can be adjusted via k-means clustering. This enables our solution to scale with data growth.
- *Accuracy:* We apply a statistical heuristic for thresholding the autoencoder reconstruction error, which enables our solution to identify performance defects with significantly higher accuracy.

In the rest of this paper, after some background, we first present our approach and then show the effectiveness of our solution with an experimental study on real-world benchmarks (PARSEC [17] and Phoenix [57] benchmark suites) and open-source software packages (Boost, Memcached, and MySQL). With only 4% average profiling overhead, our tool can successfully detect three types of performance regressions common in parallel software (true sharing, false sharing, and NUMA latency), at consistently higher accuracy than two state-of-the-art approaches [21, 33].

## 2 Motivation

Industrial software development is constantly seeking to accelerate the rate in which software is delivered. Due to the ever increasing frequency of deployments, software performance defects are leaking into production software at an alarming rate [34]. Because this trend is showing no sign of slowing, there is an increasing need for the practical adoption of techniques that automatically discover performance anomalies to prevent their integration to production-quality software [54]. To achieve this goal, we must first understand the challenges that inhibit building practical solutions. This section discusses such challenges and their potential solutions.

### 2.1   Challenges: Diagnosing Software Performance Regressions

Detailed software performance diagnostics are hard to capture. We see two core challenges.

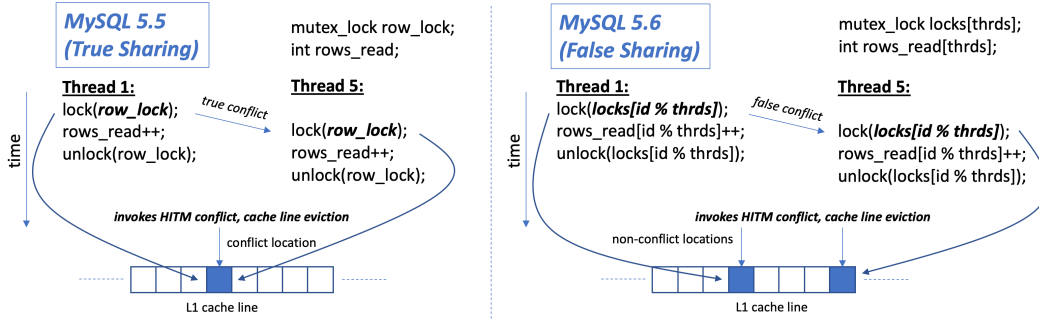

Figure 1: Example of performance regressions in parallel software.

**Examples are limited.** Software performance regressions can manifest in a variety of forms and frequencies. Due to this, it is practically impossible to exhaustively identify all of them a priori. In contrast, normal performance behaviors are significantly easier to observe and faithfully capture.

**Profiling may perturb performance behavior.** Software profiling via code instrumentation may cause perturbations in a program's run-time behavior. This is especially true for parallel software, where contention signatures can be significantly altered due to the most minute *probe effect* [26, 48] (e.g., a resource contention defect may become unobservable).

These challenges call for an approach that *(i)* does not rely on training data that includes performance regressions and *(ii)* uses a profiling technique which incurs minimal execution overhead (i.e., less than 5%) as to not perturb a program's performance signature. Next, we provide concrete examples of performance bugs that are sensitive to these two criteria.

## 2.2 Examples: Software Performance Regressions

**Cache contention** may occur when multiple threads of a program attempt to access a shared memory cache concurrently. It comes in two flavors: (i) *true sharing*, involving access to the same memory location, and (ii) *false sharing*, involving access to disjoint memory locations on the same cache line. For example, a true sharing defect in MySQL 5.5 is shown in Figure 1(a). Unfortunately, developer's attempt to fix this issue could cause a performance regression due to false sharing defect. This defect in Figure 1(b) was introduced into MySQL version 5.6, leading to more than a 67% performance degradation [9].

**NUMA latency** may arise in Non-Uniform Memory Access (NUMA) architectures due to a mismatch between where data is placed in memory vs. the CPU threads accessing it. For example, the streamcluster application of the PARSEC benchmark was shown to experience a 25.7% overall performance degradation due to NUMA [17].

These types of performance defects are generally challenging to identify from source code. An automatic approach can leverage HWPCs as a feature to identify these defects (more in Section 4.2).

## 2.3 A New Approach: Zero-Positive Learning Meets Hardware Telemetry

To address the problem, we propose a novel approach that consists of two key ingredients: zero-positive learning (ZPL) [36] and hardware telemetry [10].

ZPL is an implicitly supervised ML technique. It is a specific instance of one-class classification, where all training data lies within one class (i.e., the non-anomalous space). ZPL was originally developed for anomaly detection (AD). In AD terminology, a positive refers to an anomalous data sample, while a negative refers to a normal one, thus the name *zero-positive learning*. Any test data that sufficiently deviates from the negative distribution is deemed an anomaly. Thus, ZPL, if coupled with the right ML modeling technique, can provide a practical solution to the first challenge, as it does not require anomalous data.

Hardware telemetry enables profiling program executions using hardware performance counters (HWPCs). HWPCs are a set of special-purpose registers built into CPUs to store counts of a wide range of hardware-related activities, such as instructions executed, cycles elapsed, cache hits or misses, branch (mis)predictions, etc. Modern-day processors provide hundreds of HWPCs, and more

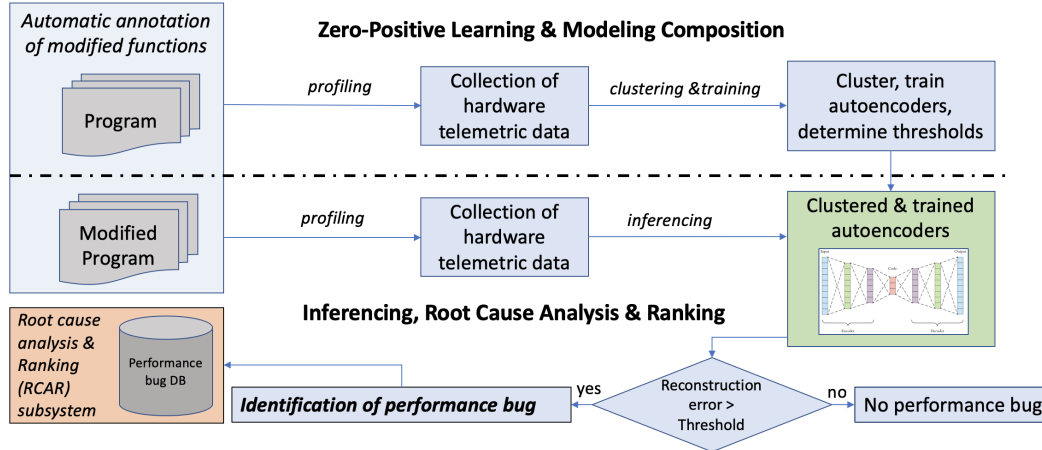

Figure 2: Overview of AutoPerf

are being added with every new architecture. As such, HWPCs provide a lightweight means for collecting fine-grained profiling information without modifying source code, addressing the second challenge.

## 3 Related Work

There has been an extensive body of prior research in software performance analysis using statistical and ML techniques [15, 31, 38, 53, 58]. Most of the past ML approaches are based on traditional supervised learning models (e.g., Bayesian networks [20, 59], Markov models [29], decision trees [33]). A rare exception is the unsupervised behavior learning (UBL) approach of Dean et al., which is based on self-organizing maps [21]. Unfortunately, UBL does not perform well beyond a limited number of input features. To the best of our knowledge, ours is the first scalable ML approach for software performance regression analysis that relies only on normal (i.e., no example of performance regression) training data.

Prior efforts commonly focus on analyzing a specific type of performance defect (e.g., false and/or true sharing cache contention [22, 33, 39–41, 51, 63], NUMA defects [42, 56, 61]). Some of these also leverage HWPCs like we do [14, 18, 28, 40, 56, 61]. However, our approach is general enough to analyze any type of performance regression based on HWPCs, including cache contention and NUMA latency. Overall, the key difference of our contribution lies in its practicality and generality. Section 5 presents an experimental comparison of our approach against two of the above approaches [21, 33].

## 4 The Zero-Positive Learning Approach

In this section, we present a high-level overview of our approach, followed by a detailed discussion of its important and novel components.

### 4.1 Design Overview

A high-level design of AutoPerf is shown in Figure 2. Given two versions of a software program, AutoPerf first compares their performance. If a degradation is observed, then the cause is likely to lie within the functions that differ in the two versions. Hence, AutoPerf automatically annotates the modified functions in both versions of the program and collects their HWPC profiles. The data collected for the older version is used for zero-positive model training, whereas the data collected for the newer version is used for inferencing based on the trained model. AutoPerf uses an autoencoder neural network to model normal performance behavior of a function [60]. To scale with a large number of functions, training data for functions with similar performance signatures are clustered together using k-means clustering and a single autoencoder model per cluster is trained [35]. Performance regressions are identified by measuring the reconstruction error that results from testing

the autoencoders with profile data from the new version of the program. If the error comes out to be sufficiently high, then the corresponding execution of the function is marked as a performance bug and its root cause is analyzed as the final step of the diagnosis.

## 4.2 Data Collection

Modern processors provide various hardware performance counters (HWPCs) to count low-level system events such as cache misses, instruction counts, memory accesses [10]. AutoPerf uses Performance Application Programming Interface (PAPI) to read values of hardware performance counters [49]. For example, for the specific hardware platform that we used in our experimental work (see Section 5.1 for details), PAPI provides access to 50 different HWPCs. Many of these performance counters reflect specific performance features of a program running on the specific hardware. For example, Hit Modified (HITM) is closely related to cache contention [45]. Essentially, this counter is incremented every time a processor accesses a memory cache line which is modified in another processor's cache. Any program with true or false sharing defects will see a significant increase in the HITM counter's value. Similarly, the counter for off-core requests served by remote DRAM (OFFCORE_RESPONSE: REMOTE_DRAM) can be used to identify NUMA-related performance defects [42]. AutoPerf exploits these known features in its final root-cause analysis step.

To collect HWPC profiles of all modified functions, we execute both of the annotated program versions with a set of test inputs (i.e., regression test cases). Test inputs generally capture a variety of different input sizes and thread counts. During each execution of an annotated function foo, AutoPerf reads HWPCs at both the entry and the exit points of foo, calculates their differential values, normalizes these values with respect to the instruction count of foo and thread count, and records the resulting values as one sample in foo's HWPC profile.

## 4.3 Diagnosing Performance Regressions

AutoPerf uses HWPC profiles to diagnose performance regressions in a modified program. First, it learns the distribution of the performance of a function based on its HWPC profile data collected from the original program. Then, it detects deviations of performance as anomalies based on the HWPC profile data collected from the modified program.

### 4.3.1 Autoencoder-based Training and Inference

Our approach to performance regression automation requires to solve a zero-positive learning task. Zero-positive learning involves a one-class training problem, where only negative (non-anomalous) samples are used at training time [50]. We employ autoencoders to learn the data distribution of the non-anomalous data [13]. At test time, we then exploit the autoencoder to discover any deviation that would indicate a sample from the positive class. The autoencoder model is a natural fit for our ZPL approach, since it is unsupervised (i.e., does not require labeled training data as in one-class training) and it works well with multi-dimensional inputs (i.e., data from multiple HWPCs).

To formalize, let $\{\mathbf{x}_i\}_{i=1}^{N_{old}}$ be a set of $N_{old}$ samples obtained from profiling the old version of the function foo. Next, we train an autoencoder $\mathcal{A}_{\text{foo}}(\mathbf{x}) = f(g(\mathbf{x}))$ such that it minimizes the reconstruction error over all samples, i.e., $\mathcal{L}(\mathbf{x}_i, \mathcal{A}_{\text{foo}}(\mathbf{x}_i)) = \sum_i \|\mathbf{x}_i - \mathcal{A}_{\text{foo}}(\mathbf{x}_i)\|_2^2$. During training, the autoencoder $\mathcal{A}_{\text{foo}}(\mathbf{x})$ learns a manifold embedding represented by its encoder $g(\mathbf{x})$. Its decoder $f(\mathbf{x})$ learns the projection back to sample space. Learning the manifold embedding is crucial to the autoencoder to reconstruct a sample with high fidelity.

Once the autoencoder is trained, AutoPerf collects an additional set of samples $\{\mathbf{z}_i\}_{i=1}^{N_{new}}$ profiling the newer version of function foo's code. Next, the system discovers anomalies by encoding and decoding the new samples $\mathbf{z}_i$ and measuring the reconstruction error, i.e.,

$$\epsilon(\mathbf{z}_i) = \|\mathbf{z}_i - \mathcal{A}_{\text{foo}}(\mathbf{z}_i)\|_2 \tag{1}$$

If the reconstruction error for a sample $\mathbf{z}_i$ is above a certain threshold $\gamma$, i.e., $\epsilon > \gamma$, the sample is marked as anomalous, as it lays sufficiently distant from its back-projected reconstruction.

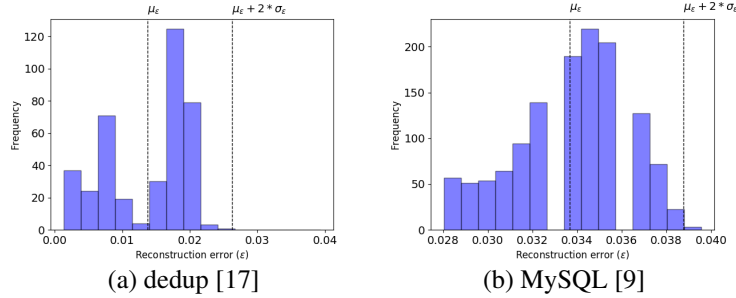

Figure 3: Histograms of reconstruction error $\epsilon$ for training samples $\{\mathbf{x}_i\}$ of two real datasets

### 4.3.2 Reconstruction Error Threshold Heuristic

The success to detect the anomalous samples heavily depends on setting the right value for threshold $\gamma$. A value too high and we may fail to detect many anomalous samples, raising the number of false negatives. A value too low, and AutoPerf will detect many non-anomalous samples as anomalous, increasing the number of false positives. Figure 3(a) and (b) show the reconstruction errors for the training samples for the dedup and MySQL datasets, respectively [9, 17]. Clearly, the difference in histograms signals that naïvely setting a threshold would not generalize across datasets or even functions.

The skewness in the reconstruction error distributions of all test applications ranges from -0.46 to 0.08. The kurtosis ranges from 1.96 to 2.64. Therefore, we approximate the reconstruction error's distribution with a Normal distribution and define a threshold $\gamma(t)$ relative to the errors as

$$\gamma(t) = \mu_\epsilon + t\sigma_\epsilon \tag{2}$$

where $\mu_\epsilon$ is the mean reconstruction error and $\sigma_\epsilon$ its standard deviation for the training samples $\{\mathbf{x}_i\}$. The $t$ parameter controls the level of thresholding. For example, with $t = 2$, the threshold provides (approximately) a 95% confidence interval for the reconstruction error.

To find the cause (and type) of performance regression, we calculate the reconstruction error (RE) for each performance counter corresponding to an anomalous sample and then, sort the counters accordingly. We take a majority vote among all performance counters for each anomalous sample. The counter that comes first is the one that causes the performance regression. We report that counter and the corresponding regression type as the root cause.

### 4.3.3 Scaling to Many Functions via k-means Clustering

So far, we have focused on analyzing the performance of a single function that is modified with new code. In reality, the number of functions that change between versions of the code is higher. For example, 27 functions are modified between two versions of MySQL used in our experiments [9]. Training one autoencoder per such function is impractical. Furthermore, the number of samples required to train these grows, too. To alleviate this, we group multiple functions into clusters and assign an autoencoder to each group. AutoPerf applies k-means clustering for this purpose [35]. It computes $k$ clusters from the training samples. Then, we assign function $f$ to cluster $c$, if $c$ contains more samples of $f$ than any other cluster. For each cluster $c$, we build one autoencoder. We train the autoencoder using the training samples of all the functions that belong to that cluster. During inferencing, when we analyze profiling samples for a newer version of a function, we feed them to the autoencoder of the cluster where that function belongs to.

## 5 Experimental Evaluation

In this section, we *(i)* evaluate AutoPerf's ability to diagnose performance regressions and compare with two state-of-the-art machine learning based approaches: Jayasena et al. [33] and UBL [21], *(ii)* analyze our clustering approach, and *(iii)* quantify profiling and training overheads.

Table 1: Diagnosis ability of AutoPerf vs. DT [33] and UBL [21]. *TS* = True Sharing, *FS* = False Sharing, and *NL* = NUMA Latency. $K$, $L$, $M$ are the # of executions ($K = 6$, $L = 10$, $M = 20$).

| Normal Program | False Positive Rate | | | Anomalous Program | Defect Type | False Negative Rate | | |
|---|---|---|---|---|---|---|---|---|
| | AutoPerf | DT | UBL | | | AutoPerf | DT | UBL |
| $\text{blackscholes}_L$ | 0.0 | N/A | 0.2 | $\text{blackscholes}_K$ | *NL* | 0.0 | N/A | 0.0 |
| $\text{bodytrack}_L$ | 0.0 | 0.7 | 0.8 | $\text{bodytrack}_K$ | *TS* | 0.0 | 0.17 | 0.1 |
| $\text{dedup}_L$ | 0.0 | 1.0 | 0.2 | $\text{dedup}_K$ | *TS* | 0.0 | 0.0 | 0.0 |
| $\text{histogram}_M$ | 0.0 | 0.0 | 0.0 | $\text{histogram}_M$ | *FS* | 0.0 | 0.1 | 1.0 |
| $\text{linear\_regression}_M$ | 0.0 | 0.3 | 0.0 | $\text{linear\_regression}_M$ | *FS* | 0.0 | 0.4 | 0.35 |
| $\text{reverse\_index}_M$ | 0.0 | 0.4 | 0.15 | $\text{reverse\_index}_M$ | *FS* | 0.0 | 0.1 | 0.05 |
| $\text{streamcluster}_L$ | 0.0 | N/A | 0.6 | $\text{streamcluster}_K$ | *NL* | 0.0 | N/A | 0.1 |
| $\text{Boost}_L$ | **0.3** | 1.0 | 0.4 | $\text{Boost}_L$ | *FS* | 0.0 | 0.2 | 0.2 |
| $\text{Memcached}_L$ | 0.0 | 1.0 | 0.4 | $\text{Memcached}_L$ | *TS* | 0.0 | 0.4 | 0.3 |
| $\text{MySQL}_L$ | **0.2** | 1.0 | 0.1 | $\text{MySQL}_L$ | *FS* | 0.0 | 0.5 | 0.8 |

## 5.1 Experimental Setup

We used PAPI to read hardware performance counter values [49], and Keras with TensorFlow to implement autoencoders [19]. PAPI provides a total of 50 individualized and composite HWPCs. We read the 33 individualized counters during profiling as input features to AutoPerf. We performed all experiments on a 12-core dual socket Intel Xeon® Scalable 8268 processor [3] with 32GB RAM. We used 7 programs with known performance defects from the PARSEC [17] and the Phoenix [57] benchmark suites. Additionally, we evaluated 3 open-source programs: Boost [2], Memcached [4], and MySQL [5].

## 5.2 Diagnosis Ability

We experiment with 10 programs to evaluate AutoPerf. Two versions of source code for each program are used for these experiments: 1) a version without any performance defect; 2) a version where a performance defect is introduced after updating one or more functions in the first version. We run the first version $n$ number of times. If a system reports $x$ number of these runs as anomalous (i.e., positive), we define *false positive rate* as $x/n$. Similarly, we run the second version $m$ number of times and define *false negative rate* as $x/m$, where $x$ is the number of anomalous runs detected as non-anomalous. Each run of a program uses different inputs.

AutoPerf's diagnosis results are summarized in Table 1. We experimented with 3 different types of performance defects across 7 benchmark programs and 3 real-world applications. These are known performance bugs (confirmed by developers) in real-world and benchmark applications. A version of each application, for which corresponding performance defect is reported, is used for generating anomalous runs. AutoPerf detects performance defects in all anomalous runs. However, it reports 3 false positive runs in Boost and 2 false positive runs in MySQL. Anomalies in Boost are detected in a function that implements a *spinlock*. It implements lock acquisition by iteratively trying to acquire the lock within a loop. Moreover, these runs are configured with increased number of threads. We suspect that these false positive test runs experienced increased lock contention, which was not present in training runs. This could be improved by increasing the variability of inputs for training runs. The two false positive runs in MySQL are reported in two functions. These are small functions with reduced number of instructions, which could affect the accuracy of profiling at a fixed sampling rate.

We quantitatively compared AutoPerf with two state-of-the-art machine learning based approaches: Jayasena et al. [33] and UBL [21]. Jayasena et al. uses a decision tree of 12 performance counters to detect true sharing and false sharing defects (DT in Table 1). This approach is limited to detection of false sharing and true sharing types of performance defect. Therefore, it cannot detect the NUMA performance defects in blackscholes and streamcluster. Moreover, [33] uses a fixed ratio of various counters and therefore, cannot detect all anomalous runs in 6 programs and reports false positive runs for all 8 programs.

We implemented UBL using a $120 \times 120$ self-organizing map (SOM) to detect performance anomalies. Table 1 shows UBL reports greater number of false positive runs for 7 programs and greater false negative runs for 7 programs. The reduction in accuracy is caused by SOM's limitation in handling large variations in performance counter values. Overall, AutoPerf produces false positives for Boost and MySQL, whereas other approaches produces false positives or false negatives nearly for every program. We further evaluated the anomaly prediction accuracy of AutoPerf using the standard receiver operating characteristic (ROC) curves. Figure 4 shows ROC curves for Boost and MySQL. Although AutoPerf produces false positives for these two applications, the ROC curves show that it achieves better accuracy than UBL for these two applications.

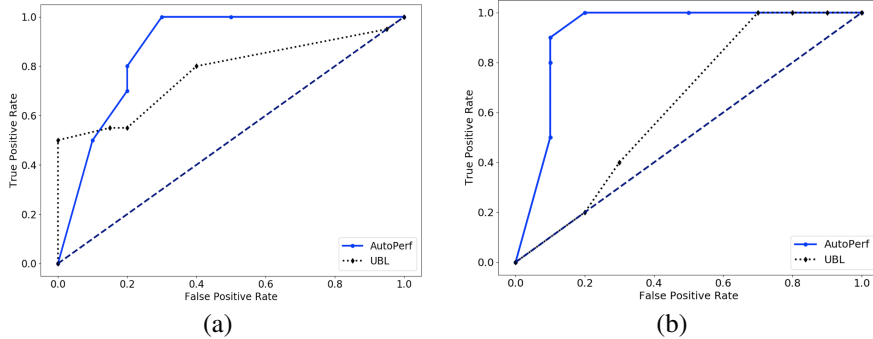

(a)                                                    (b)

Figure 4: Diagnosis of false sharing defects in (a) Boost and (b) MySQL. True positive rates and false positive rates of AutoPerf and state-of-the-art approach UBL [21] for an application with different thresholds are shown in each figure.

## 5.3   Impact of Clustering

To analyze many functions that change between versions of code with reduced number of autoencoders, AutoPerf combines groups of similar functions into clusters and train an autoencoder for each cluster. We experimented with AutoPerf's accuracy to evaluate if the clustering reduces the accuracy of the system compared to using one autoencoder for each function.

One way to evaluate this is to test it against a program with multiple performance defects in different functions. To achieve this, we performed a sensitivity analysis using a synthetic program constructed with seven functions. We created a modified version of this program by introducing performance defects in each of these functions and evaluated the $F_1$ score of AutoPerf with different number of clusters for these seven functions in the program. AutoPerf achieves a reasonable $F_1$ score (from 0.73 to 0.81) using one autoencoder per function. When it uses one autoencoder across all seven functions, $F_1$ degrades significantly to 0.31. Using k-means clustering we can achieve reasonable accuracy even without one autoencoder per function. As shown in Figure 5(a), there is an increase in accuracy ($F_1$ score) as $k$ increases from 2 to 3 to 4.

We evaluate the effects of clustering in three real-world programs: Boost, Memcached, and MySQL. Figure 5(b) shows accuracy of these programs using $F_1$ score. For Memcached, AutoPerf creates three clusters from eight candidate functions (i.e., changed functions). The $F_1$ score after clustering becomes equal to the $F_1$ score of an approach that uses one autoencoder per function. For other two programs: Boost and MySQL, clustering results in slightly reduced $F_1$ score. However, as shown in Figure 5(c), the clustering approach reduces overall training time of AutoPerf by 2.5x to 5x.

## 5.4   Effectiveness of the Error Threshold

We evaluated the effectiveness of our threshold method for $\gamma(t)$. We compared with a base approach of setting an arbitrary threshold based on the input vector $x$ instead of reconstruction errors. This arbitrary threshold, $\alpha(t)$, implies that if the difference between the output and input vector length is more than $t\%$ of the input vector length $x$, it is marked as anomalous. We compared accuracy of AutoPerf with UBL and this base approach using the mean true positive rates and mean false positive rates of these approaches across 10 candidate applications listed in Table 1. Figure 6(a) shows

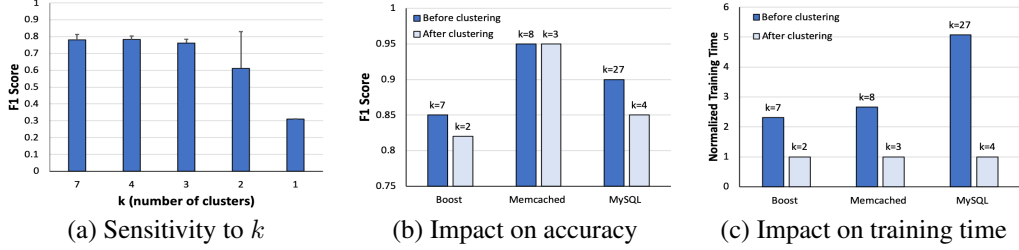

| (a) Sensitivity to $k$ | (b) Impact on accuracy | (c) Impact on training time |

Figure 5: Impact of clustering, where $k$ denotes the number of clusters (i.e., autoencoders)

the accuracy of AutoPerf using arbitrary threshold and $\gamma(t)$. We evaluated AutoPerf with different thresholds determined using equation (2), where values of $t$ ranges from 0 to 3. AutoPerf achieves true positive rate of 1 and false positive rate of 0.05 using $\gamma(t)$ at $t = 2$. For arbitrary threshold using $\alpha(t)$, we experimented with increasing values of t from 0 to 55, at which point both true positive rate and false positive rate become 1. Figure 6 also shows the accuracy of UBL with different thresholds. $\gamma(t)$ achieves increased accuracy compared to UBL and $\alpha(t)$. Moreover, $\alpha(t)$ performs even worse than the best results from UBL.

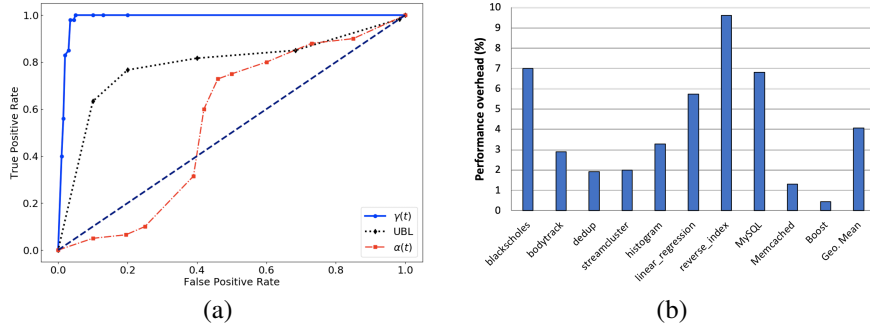

| (a) | (b) |

Figure 6: (a) Effect of error threshold, (b) Profiling overhead.

## 5.5 Profiling and Training Overheads

Profiling of a program introduces performance overhead. However, AutoPerf uses HWPCs to implement a lightweight profiler. The execution time of an application increases by only 4%, on average, with AutoPerf. MySQL experiments results in the highest performance overhead of 7% among three real-world applications. AutoPerf monitors greater number of modified functions in MySQL compared to the other two real-world applications: Memcached and Boost. We also collected the training time of autoencoders. On average, it takes approximately 84 minutes to train an autoencoder. An autoencoder for MySQL, which models a cluster with many functions, takes the longest training time, which is little less than 5 hours using our experimental setup (Section 5.1).

## 6 Conclusion

In this paper, we presented AutoPerf, a generalized software performance analysis system. For learning, it uses a fusion of zero-positive learning, k-means clustering, and autoencoders. For features, it uses hardware telemetry in the form of hardware performance counters (HWPCs). We showed that this design can effectively diagnose some of the most complex software performance bugs, like those hidden in parallel programs. Although HWPCs are useful to detect performance defects with minimal perturbation, it can be challenging to identify the root cause of such bugs with HWPCs alone. Further investigation into a more expressive program abstraction, coupled with our zero-positive learning approach, could pave the way for better root cause analysis. With better root cause analysis, we might be able to realize an automatic defect correction system for such bugs.

**Acknowledgments**

We thank Jeff Hammond for his suggestions regarding experimental setup details. We thank Mostofa Patwary for research ideas in the early stages of this work. We thank Pradeep Dubey for general research guidance and continuous feedback. We thank Marcos Carranza and Dario Oliver for their help improving the technical correctness of the paper. We also thank all the anonymous reviewers and area chairs for their excellent feedback and suggestions that have helped us improve this work.

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
