[Supplementary Material · supplementary.pdf]

# 1 Details of Performance Regressions

## Benchmark Programs

AutoPerf diagnosed true sharing defects bodytrack and dedup programs of Parsec suite We used native inputs for both programs. Bodytrack implements a work stealing algorithm using `TicketDispenser`. Every thread updates `value` in `getTicket()` and suffers from a true sharing cache contention defect. The defect degrades performance by $\approx 10\%$. We resolved this by statically dividing the workload equally among the threads. Dedup has a number of contented locks (`ht_lock`) to guard a hash table. These locks are accessed by multiple threads and introduce cache contention in the form of true sharing. We reduced contention in `ht_lock` using a different hash function. [1] We tested AutoPerf with 10 non-anomalous and 6 anomalous runs of each of these programs. However, HITM is ranked 2 for bodytrack. The counter for L2 store misses (L2_STM) is reported as the top ranked one.

AutoPerf diagnosed NUMA performance defects in two programs from Parsec: blackscholes and streamcluster. In streamcluster, a single array, called `block`, is read and written by many threads. Because the array is allocated on a single node, the program suffers from significant remote memory access latency [4]. We created a non-anomalous version of this program by interleaving the memory allocated for this array. A similar NUMA performance defect, associated with an array, `buffer`, exists in blackscholes [7]. This defect causes a 7.5% performance degradation. AutoPerf diagnosed both NUMA performance defects with a perfect $F_1$ score and *OFFCORE_RESPONSE: REMOTE_DRAM* ranked as top counter.

AutoPerf detected false sharing defects in three programs of Phoenix benchmark suite: histogram, linear_regression, and reverse_index. All defects were previously reported in prior work [5, 6]. AutoPerf detected these false sharing defects with a perfect $F_1$ score and HITM ranked as the top counter.

## Open Source Applications

AutoPerf diagnosed performance defects in three open source programs: Boost library, Memcached, and MySQL.

Boost provides portable C++ libraries. A *spinlock* (a type of lock) is implemented as a pool of size 41 in version-1.38.0. Different threads may access different spinlocks. Each spinlock is 4 bytes long and, therefore, multiple spinlocks can be located in a single cache line. This results in a false sharing cache contention defect. This defect degrades Boost's performance by up to 30% [2]. We fixed the issue by adding necessary padding. We tested 10 non-anomalous and 10 anomalous runs of Boost using AutoPerf. AutoPerf can diagnose false sharing in Boost library in 10 anomalous runs, but falsely reports in 3 non-anomalous runs as defects. So, $F_1$ score becomes 0.82.

Memcached is a distributed memory object caching system. AutoPerf detected false sharing defects in `do_item_alloc` function of version 1.4.10. This function is changed from version 1.4.4 to 1.4.10 by introducing `cache_lock` shared variable. Multiple threads accessing this variable can create significant true sharing contention. We used memslap [1] benchmark for this. AutoPerf detected cache contention defects in Memcached with no false positive or negative.

In MySQL, AutoPerf detected false sharing involving the `n_rows_read` variable. This variable was changed to an array from version 5.5 to 5.6. We used sysbench OLTP benchmarks.

# 2 Source Code and Data

Source code of AutoPerf is uploaded to github: `https://github.com/mejbah/AutoPerf/tree/master/autoperf`.
Dataset used for our experiments can be accessed from `https://drive.google.com/drive/folders/17Hz-OmQ0W4uxC_7E393DJzrRl3EQ4SXJ?usp=sharing`

## Footnotes

[1] Both of these bugs were discovered in prior work [8, 3].

[5] LIU, T., AND BERGER, E. D. Sheriff: Precise detection and automatic mitigation of false sharing. In *Proceedings of the 2011 ACM International Conference on Object Oriented Programming Systems Languages and Applications* (New York, NY, USA, 2011), OOPSLA '11, ACM, pp. 3–18.

[6] LIU, T., TIAN, C., HU, Z., AND BERGER, E. D. PREDATOR: Predictive False Sharing Detection. In *Proceedings of the 19th ACM SIGPLAN Symposium on Principles and Practice of Parallel Programming* (New York, NY, USA, 2014), PPoPP '14, ACM, pp. 3–14.

[7] LIU, X., AND MELLOR-CRUMMEY, J. A tool to analyze the performance of multithreaded programs on numa architectures. In *Proceedings of the 19th ACM SIGPLAN Symposium on Principles and Practice of Parallel Programming* (New York, NY, USA, 2014), PPoPP '14, ACM, pp. 259–272.

[8] LUO, L., SRIRAMAN, A., FUGATE, B., HU, S., POKAM, G., NEWBURN, C. J., AND DEVIETTI, J. LASER: Light, Accurate Sharing dEtection and Repair. In *2016 IEEE International Symposium on High Performance Computer Architecture (HPCA)* (March 2016), pp. 261–273.