[Reviews · NeurIPS 2019]

Reviewer 1



(Continuing my answer to "contributions"... I ran out of space) STRONG POINTS/CONTRIBUTIONS 1) The false positive rates and false negative rates observed when using AutoPerf are impressively low. 2) The paper is very well written and easy to follow. 3) The evaluation uses a wide variety of significant, real-world codes. NEGATIVE POINTS 1) The paper lacks a lot of technical depth and novelty… autoencoders for anomaly detection are widely used, and the problem domain (detecting performance bugs) has been studied previously as well. 2) It strikes me that a version of AutoPerf that was production quality could not ignore commit logs, as the authors seem to have done. Knowing what was changed in the code between P_i and P_i+1 could be very, very helpful. 3) The idea of using k-means clustering can speed things a bit, but it is perhaps not the most natural way to do this. Instead, why not just conditions the identity of the function? 4) The evaluation focuses on finding synthetic anomies, rather than real, known performance bugs. DETAILED COMMENTS One comment is that I’m not sure it makes a lot of sense to train separate autoencoders for each function (or group of functions, if you are doing the k-means thing). Likely, there are going to be certain characteristics of the distributions that are shares across all functions, and I worry that you are wasting a lot of compute power by relearning everything. One simple idea would be to learn a single autoencoder, but you would condition the encoder and decoder on the function. That is, you have a separate input to both your encoder and decoder networks that is the identity of the function, but you are conditioning on this (you are not trying to reconstruct the function label), This may allow you to get rid of the multiple autoencoders entirely, and use only one of them. It is interesting that there is this implicit assumption that any changes in performance between P_i and P_i+1 are bad… that they represent depredations in performance. Clearly, this is not the case. Someone can fix a performance bug across versions. Currently, you are likely to flag such a fix as a bug. Is there an easy way to differentiate between good and bad changes? Can this be done without having training data? One big concern that I have about this paper is technical depth and novelty. When it comes right down to it, the authors are simply proposing to use a state-of-the-art approach to anomaly detection to the problem of finding changes in performance across multiple versions of a program. It appears that using anomaly detection to fund performance problems is not new or novel, and I know enough about anomaly detection to know that sing an autoencoder + reconstruction error to do anomaly detection is a fairly standard thing to do nowadays. So what we are left with is a solidly-written paper that applies a standard anomaly detection methodology to a problem that has been studied previously. For this reason, it is tough to get too excited about the paper. It would be nice to know how many functions are in each of the tested codes. I just assumed that a lot of functions were tested for performance bugs, but according to Figure 5, this may not be the case. In fact, I’m surprised that Boost, for example has k = 7 before clustering. Does this mean that Boost has 7 functions?!?! This is very surprising, since Boost is huge. Obviously, there is something that I am missing here, or else maybe the authors have radically down-sampled the codes for evaluation. Regardless, this needs to be explained. It would be good to have more detailed about the anomalies introduced. Note using val anomalies weakens the experimental evaluation, but not having detailed on exactly how the anomies were introduced weakens it even more.

Reviewer 2



= Originality This is an application paper, proposing a new model/task combination. I am not an expert in the area, but the application of using zero-positive learning with autoencoders to such low-level hardware instrumentation data was an interesting new idea to me. = Clarity The paper is very well-written, and after reading it, I feel that I understood the approach well and could reproduce it. = Quality The article presents a basic idea together with a number of practical extensions (such as the clustering of functions) that make it more applicable. The experiments support most claims, though the number of examples and example runs are low, and choices there are unexplained: Why were 4 four benchmark programs (blackscholes, bodytrack, dedup, streamcluster) run more often for the "normal" version of the program than for the anomalous version? [The authors replied to this saying "Performance anomalies in some applications do not occur for all test cases". I do not understand how that changes the number of executions: Shouldn't the number of test cases actually executed be constant across the two versions of the program?] The phrasing of the metrics could also be misunderstood, as the "false positive" rate is often interpreted as the ratio of wrongly generated warnings in program analysis, whereas here, it related to the overall number of analysed runs. This may be problematic because the system is likely to analyse "normal" programs far more often than those with performance regressions, and so the rate of wrongly reported runs may still be very high. I am also somewhat surprised that considering differing functions alone is sufficient, as many performance regressions stem from a mismatch between allocation and access patterns, but the performance will only be measured (and hence noticed as degreaded) at access time. Hence, in cases where changes to data allocation causes a slowdown at access time, the heuristic of only considering changed functions can easily fail. [The authors replied to this explaining that additional analyses could identify functions that could be impacted by changes; however, I believe that the reply does not consider the implication that such heuristics would mean that much more code would need to be analysed, making it harder to find anomalies] It would be helpful if the authors could comment on these issues in their feedback, and provide some insight if they were observed in the benchmarks. = Significance I believe this submission has little significance for the machine learning community, as it does not put forth a new research direction or insights into ML. If the authors indeed release their implementation and dataset, other researchers may also investigate this task, but I feel that a submission to a conference of the software engineering or performance analysis communities may be more impactful. However, I understand the pain of getting cross-discipline work published, and believe that NeurIPS would be a suitable venue for this work.

Reviewer 3



--Summary-- This work introduces AutoPerf, a tool for software performance monitoring. This approach attempts to detect regressions (performance bugs) in software by examining changes in hardware telemetry data. In particular, they examine HPCs (Hardware Performace Counters) which contain metrics about low level system events such as cache misses and instruction counts. This can be useful for detecting performance bugs encountered in parallel programming scenarios such as cache contention. The authors train an autoencoder on HPC samples from a non-buggy program and compare the reconstruction error against samples from a candidate program in order to detect whether the candidate program introduces a performance regression. Since the number of functions can be large, the authors suggest clustering similar functions using k-means. The authors evaluate their method across 3 types of regressions, 10 performance bugs and 7 datasets and show that their method outperforms competing methods. --Strengths-- - This is a well written paper that makes an important contribution to an emerging field of ML for systems. - The authors do careful analysis and present a strong case for the value of their method. - This method is likely to be immediately useful as a tool for catching performance regressions in software. --Quality-- Overall this paper is of good research quality and the authors have undertaken careful analysis. The authors can strengthen their paper if they address the following issues: - From a modeling perspective, the authors need to justify why they chose an autoencoder. The main argument put forward by the authors is that supervised data is not available. However, the authors did not consider other unsupervised density modeling techniques such as Gaussian Mixture Models (GMMs) or Variational Auto Encoders (VAEs). These models can be used to directly evaluate the likelihood of test data instead of going through a reconstruction step. - Reconstruction error may not be Gaussian distributed. Looking at figure 3(a) it seems like the distribution is somewhat skewed. The authors may need to study the skew and kurtosis of reconstruction errors in order to strengthen this claim. Alternatively, if they modeled the density directly (for instance using a VAE), then they could apply the “likelihood ratio test” in order to pick deviants. - K-means clustering for scaling up to many functions has tradeoffs. For e.g. Sec 4.3.3 mentions that a function is assigned to a cluster if a majority of its HPC signatures belong to it. This is heavily context dependant. Suppose a test scales O(n) in input size, then differently sized tests will belong to different clusters. - The authors should provide end-to-end timing analysis of their method. Right now it takes 84 minutes for autoencoder training, precluding testing at presubmit time as part of a continuous integration setup. Alternate models such as GMMs can be trained faster. - Sec 5.2 describe how regressions were introduced in the code. However no commentary is made regarding the interestingness of the regression. The authors should specify if they took efforts to break the code in varying and interesting ways. - Minor comment - performance enhancement might be flagged as a bug as well since it causes deviation from expected behavior. Please clarify that deviation is one-sided. --Clarity-- The paper is well written and was easy to follow. A few minor comments: L231 replace x with y for false negative rate. The variable x is overloaded. --Reproducibility-- The authors can aid reproducibility by providing the following (if necessary, in a supplementary section): Details of using PAPI and all the HPCs considered. Details about the hyperparameters used for training their autoencoders. The training procedure and how they split their data for training and validation. --Clarifications/comments-- - The claim in Sec 1, P3 “...AutoPerf, can learn to diagnose potentially any type of regression…” is perhaps too strong. There can be various causes for regressions many of which are not fine-grained enough to be captured by HPCs. - The authors need to address limitations of their method and list causes of false alarms for a candidate program such as (1) inherent randomness (2) unrelated confounding factors. As well as false negatives - AutoPerf is only as strong as the tests. - It’s not clear from the paper, the dimensionality of input to the autoencoder. Does it take in metrics from all 50 HPCs in a vector? If not, then cross-dimensional correlations will be lost.

Reviewer 4



See contributions.

[Author Response · NeurIPS 2019]

**Rebuttal 6217:** We thank the reviewers for their feedback. Due to limited space, we couldn't address all comments. All code/experiments will be included in appendices. We use this notation for brevity: **Reviewer 1, 2, and 3 (R1, R2, R3)**.

**R1#1 - Technical Depth and Novelty:** We agree autoencoders (AEs) used for anomaly detection (AD) is not novel. We believe the novelty of our system is in how we combine multiple techniques from systems and ML into a new state-of-the-art (SOTA) solution as shown empirically. Although widely studied, to our knowledge, the software engineering (SE) community lacks such systems+ML software performance AD systems. Compared to the two prior SOTA works we analyzed, AutoPerf demonstrates that systems+ML tools may offer a path to more effective solutions.

**R1#2 - Analyzing Code Changes/Commit Logs:** We are slightly confused by your comment. If you are referring to using commit log comments (or other comments) to guide AutoPerf, there is evidence that such data can be erroneous (39% false positive rate) [iComment, SOSP07]. If you are referring to using actual changes in code, AutoPerf does, in fact, use such information. It just performs the analysis at the function-level rather than the statement-level.

**R1#3 - Synthetic vs. Real Bugs:** The performance bugs reported in Table 1 are *all* known performance bugs (confirmed by developers) in real-world and benchmark applications. None are synthetically generated (more details in R3#4).

**R1#4 - Performance Mutations (Good vs. Bad Changes):** AutoPerf provides performance diagnostics, via hardware performance counters, which identifies performance improvements and eliminates them from regression candidacy. These false positives are also found and eliminated by comparing wall-clock execution time during regression tests.

**R1#5 - Number of Functions Tested:** As regression tests generally only focus on modified code, AutoPerf only analyzes the functions that have been modified between two candidate versions of a program. Therefore, the total number of functions analyzed is a byproduct of the total number of modified functions for the analysis we performed. In our case, the number of modified functions ranged from 3 to 27 out of over +1000s, as you correctly observed (e.g., Boost).

**R2#1 - Examples and Executions:** The number of application executions is equal to the number of available relevant regression tests. The number of examples and example executions are, therefore, a byproduct of the test suite for each application. Performance anomalies in some applications do not occur for all test cases. Therefore, the number of anomalous runs are reduced for these applications.

**R2#2 - False Positive (FP) & Negative (FN) Rate:** In our experiments, AutoPerf produces zero FNs. We believe this behavior is critical for a production-quality software performance bug detection tool. This is because if even one FN (an overlooked performance bug) is deployed, it could have catastrophic quality of service or life-threatening consequences (e.g., real-time critical systems). However, FNs and FPs are directly correlated. In the literature, FPs tend to increase as FNs decrease and vice versa. Given that AutoPerf has a 0% FN rate, we believe a 20-30% FP rate is reasonable, is a notable improvement over prior SOTA, and is the preferred behavior FN/FP trade-off for this type of system.

**R2#3 - Considering Only Changed Functions:** While we didn't observe any performance bugs in unchanged functions caused by code changes elsewhere, we acknowledge this is a valid scenario. Yet, we believe such cases could be readily identified via data dependency, control flow, and call graph analysis into the changed functions.

**R3#1 - Comparison with Unsupervised Density Modeling Techniques:** AutoPerf's AE design has notably improved accuracy over the prior SOTA in detecting complex, real performance bugs in real-world applications. We also tested a GMM approach to identify performance regressions in MySQL and compared it against AutoPerf for Area Under ROC Curve (AUC). Our results illustrate that GMM has significantly reduced accuracy for this application. We would be happy to include and expand our GMM comparison for all of the applications in the camera-ready version of the paper.

**R3#2 - Distribution of Reconstruction Errors:** The skewness in the reconstruction error distributions of all test applications ranges from -0.46 to 0.08. The kurtosis ranges from 1.96 to 2.64.

**R3#3 - End-to-End Timing Analysis:** End-to-end timing includes time spent in regression testing, data collection, and inference of AutoPerf's AEs. Amongst these, data collection and inference time are overheads added by AutoPerf. However, by using HPCs we significantly reduce data collection computational overhead, which is shown in Figure-6(b). We found that the inference time of AutoPerf's AEs is negligible in the overall end-to-end time.

**R3#4 - Importance of Types of Tested Regressions:** We only experimented with real-world performance anomalies in multithreaded programs. We did not inject any bugs artificially. Moreover, these types of bugs can be notoriously challenging to detect and root-cause, even for expert programmers, and their negative impact can be catastrophic. For example, the false sharing performance regression in MySQL was introduced by an expert parallel programmer attempting (and failing) to fix a lock contention issue. This single performance bug introduced 6x performance reduction in MySQL for a common code path. We will add more details about all regression cases in our final version.

**R3#5 - Limitations:** AutoPerf's accuracy is proportional to the availability and distribution of regression testing data. It isn't designed to find performance bugs that can't be captured by HPCs. We'll tone down the strength of this claim.

[Meta-Review · NeurIPS 2019]

This paper describes a system for detecting the source of performance regressions in source code. The idea is to measure performance counters (HPCs) at a per-function level of the code, and then when a performance regression is detected, it is localized by looking for the function with most anomalous performance counters. The anomaly detection is done by training autoencoders on the HPCs, and there is a further idea to cluster functions with similar behavior profiles to avoid the need for learning an autoencoder for every function in a large code base. This is a controversial paper because there is little methodological novelty. R1 gave the lowest score and asks whether we want to allow this kind of paper in NeurIPS, worrying that if we accept any application of ML, then NeurIPS risks becoming too broad. R3 gave the highest initial score and finds the paper of high quality. R2 also supports acceptance but agrees with the lack of methodological novelty. None of the reviewers was an expert on the application, so I solicited an extra review from an expert in the application area. The expert's opinion is that the paper presents a pretty good problem formulation and would be a good foundation for future ML research (though there is a simplifying assumption in the work that doesn't hold for all performance regressions). It's worth considering R1's question. For me (AC), the reason to prefer this paper to the average ICSE ML paper is that it introduces a problem that I've never seen in an ML venue, uses an interesting source of data that will be new to ML people (HPCs), follows reasonable modern ML practices (autoencoders for anomaly detection), would benefit from more advanced ML, and it could get ML people excited about the problem (the paper is easy to ready for an ML audience). I agree that we don't want NeurIPS to only have this kind of paper, but having a mix of methodological, application, and infrastructure (e.g., software toolkits, datasets) papers is healthy IMHO.